# Construction of a Wood Nanofiber–Bismuth Halide Photocatalyst and Catalytic Degradation Performance of Tetracycline from Aqueous Solutions

**DOI:** 10.3390/molecules29143253

**Published:** 2024-07-10

**Authors:** Jiarong She, Cuihua Tian, Yan Qing, Yiqiang Wu

**Affiliations:** 1College of Materials Science and Technology, Central South University of Forestry and Technology, Changsha 410004, China; shejiarong@163.com (J.S.); tian1990c@126.com (C.T.); qingyan0429@163.com (Y.Q.); 2Hunan Academy of Forestry, Changsha 410018, China

**Keywords:** BiOBr, wooden nanofibers, tetracycline, photocatalytic degradation

## Abstract

Nanostructured bismuth oxide bromide (BiOBr) has attracted considerable attention as a visible light catalyst. However, its photocatalytic degradation efficiency is limited by its low specific surface area. In this study, a solvothermal approach was employed to synthesize BiOBr, which was subsequently loaded onto cellulose nanofibers (CNFs) to obtain a bismuth halide composite catalyst. The performance of this catalyst in the removal of refractory organic pollutants such as tetracycline (TC) from solutions under visible light excitation was examined. Our results indicate that BiOBr/CNF effectively removes TC from the solution under light conditions. At a catalyst dosage of 100 mg/L, the removal efficiency for TC (with an initial concentration of 100 mg/L) was 94.2%. This study elucidates the relationship between the microstructure of BiOBr/CNF composite catalysts and their improved photocatalytic activity, offering a new method for effectively removing pollutants from water.

## 1. Introduction

Discharging various pollutants from factories and industries substantially contributes to water contamination, directly affecting human life and considerably disrupting the environmental ecosystem. In recent years, antibiotics have emerged as some of the most extensively used drugs worldwide [1]. However, antibiotics that have not been completely metabolized are primarily released directly into the environment as metabolites. Currently, high levels of antibiotic-resistant contaminants have been identified in the environment, with most of them retaining a stable active state [2,3]. Tetracycline (TC) is a typical antibiotic that exhibits minimal decomposition under natural conditions because of its broad-spectrum antibacterial properties and chemical stability, thereby posing a major threat to aquatic and terrestrial ecosystems [4,5]. To effectively address water pollution issues stemming from antibiotics, photocatalytic degradation technology based on semiconductor materials is widely used. This technology harnesses light energy to convert it into chemical energy, thereby mineralizing organic pollutants into harmless products [6,7]. Previous studies on photocatalysis have primarily focused on semiconductor materials such as metal oxides and sulfides [8,9]. Owing to its affordability, nontoxicity, and robust stability, TiO_2_ has become the most extensively investigated photocatalyst. However, the wide bandgap of TiO_2_ (~3.20 eV) renders it capable of responding to only 5.0% of ultraviolet (UV) light, thereby limiting the applicability of TiO_2_ semiconductors [10,11].

Bismuth-based semiconductors are extensively used in photocatalysis owing to their adjustable bandgap widths, controllable nanostructures, and simple synthesis procedures [12,13]. Bi can combine with various elements to form semiconductor photocatalysts and generate crystal structures with diverse energy bands, including Bi_j_X_k_ (X: O and S) [14,15], BiOX (X: Cl, Br, and I) [16], and Bi_l_X_m_O_n_ (X: W, Mo, and V) [17,18]. The layered structure of BiOX provides ample space for polarizing the associated atoms and orbitals, facilitating the formation of an internal electric field (IEF) between [Bi_2_O_2_] and halogen slabs. The induced IEF can accelerate the separation and migration of photoexcited electron–hole pairs, thereby enhancing the photocatalytic activity of BiOX [19]. Bismuth oxide bromide (BiOBr) is a highly regarded semiconductor photocatalyst that has been extensively investigated because of its chemical stability, nontoxicity, and excellent photocatalytic activity [20]. BiOBr demonstrates remarkable degradation ability against antibiotics such as TC under visible (vis) light irradiation [21]. However, the photocatalytic efficiency of BiOBr is relatively low with respect to that required for practical applications. The preparation of composite materials is an effective method for enhancing the photocatalytic activity of semiconductors. Various composites of BiOBr and carbonaceous materials, including BiOBr/g-C_3_N_4_ [22], BiOBr/graphene [23], and BiOBr/CQDs [24], have been developed and employed to improve charge transfer and separation for enhanced photocatalytic activity. Cellulose is susceptible to functionalization because of its surface abundance of hydrophilic OH groups and has notable advantages such as robust mechanical properties and a diverse array of sources. Nanotechnology, employing biodegradable cellulose as a fine framework for inorganic fillers, is poised for considerable growth [25]. The combination of inorganic nanoparticles and cellulose facilitates the uniform dispersion of nanomaterials, thereby ensuring minimal secondary pollution during water treatment. This combination also simplifies the process of separation and recovery after complete water pollution treatment [26].

For the past few years, cellulose-based photocatalytic materials have been widely used in wastewater treatment fields, including dye degradation, heavy metal adsorption, and lotion separation [27,28,29]. The high adsorption ability of photocatalysts can enhance their photocatalytic degradation of organic pollutants because the photocatalytic reactions primarily occur on the surface [30]. Compared with halogenated oxides, nanocellulose has a higher specific surface area and stronger pollutant adsorption capacity. Therefore, the combination of BiOBr and cellulose in composites can promote the synergistic adsorption and photodegradation of TC. In this study, BiOBr and cellulose nanofiber (BiOBr/CNF) composites were synthesized using a mechanical lapping approach. The lamellar BiOBr was coated on the surface of CNF with a shell structure, which realized the highly uniform immobilization of BiOBr and solved the problem of easy agglomeration of BiOBr nanopowders. Subsequently, these composites were employed to remove TC from water through adsorption and photocatalysis under light irradiation. The primary objectives of this study are as follows: (1) synthesize BiOBr/CNF composites using a simple ball milling approach; (2) characterize the changes in phase structure, microstructure, and optical properties of the composite photocatalysts resulting from ball milling; (3) determine the synergy between adsorption and photodegradation and its effect on TC removal; and (4) reveal the microscopic mechanisms of electron transfer to free radical production and the molecular conversion pathways of TC.

## 2. Results

### 2.1. Characterization of BiOBr/CNF Composite Materials

The crystallinities of CNF, BiOBr, and the composite photocatalytic materials were characterized using X-ray diffraction (XRD) patterns (Figure 1a). CNF exhibited two distinct main-intensity peaks at 15.4° and 22.6°, which is attributed to type I cellulose. The main peaks observed at 12.2°, 26.5°, 31.7°, 32.8°, 34.1°, and 59.2° correspond to the (001), (101), (102), (110), (200), and (212) planes of BiOBr, respectively, according to the JCPDS No. 09–0393. In the BiOBr/CNF composite, the diffraction peaks of BiOBr are distinguishable, whereas the carbon peak is less prominent. This is because the carbon peaks are masked by the BiOBr peaks [31]. The surface functional groups of the samples were identified using Fourier transform infrared spectroscopy (FTIR). In Figure 1b, the peaks observed at 1383 cm^−1^ and 3477 cm^−1^ originated from the stretching vibration of O−H groups, which may be attributed to the H_2_O molecules absorbed on the BiOBr surface. The peak observed at 1622 cm^−1^ is attributed to the Bi−Br bond. The peak at 516 cm^−1^ originated from the stretching vibration of the Bi−O bond [32]. The presence of these characteristic peaks confirms the successful preparation of the BiOBr/CNF composite.

The scanning electron microscopy (SEM) images (Figure 2a,b) show the morphologies of BiOBr/CNF. As can be seen in the images, lignin exhibits good depolymerization and the surface of the treated wood fibers is relatively smooth. This indicates the removal of substrates such as silica and pectin, which enables the exposure of a larger specific surface area for loading active particles. As illustrated in Figure 2b, the BiOBr nanosheets are uniformly distributed across the surface of the CNF material, and the average size of the BiOBr is ~174.26 nm, as calculated from Figure 2b. To further examine the microstructure of the BiOBr/CNF composite, transmission electron microscopy (TEM) was used. As shown in Figure 3a,b, the BiOBr nanosheets are closely distributed on the CNF. These findings confirm that the BiOBr nanosheets strongly adhere to the CNF, forming a tight interfacial contact. This interfacial contact facilitates electron migration at the contact interface of the materials.

As the light absorption properties of catalysts are important factors in determining light response [33,34], the UV–vis diffuse reflection (UV–vis DRS) spectrum of CNF, BiOBr, and BiOBr/CNF are characterized in Figure 4. It can be found that the CNF has a strong absorption intensity in UV light, and the corresponding absorption edge is determined to be 269 nm. Due to the fact that CNF cannot be excited as a photocatalyst by visible light, so it may play a role as a photosensitizer in this study. For BiOBr, the light adsorption range is 200 nm~445 nm, indicating that BiOBr has a response ability in visible light. Compared with BiOBr, the light adsorption intensity from UV to Vis spectrum is significantly enhanced, which means that the coupling of BiOBr with CNF could effectively improve light absorption and boost the photocatalytic activity. Furthermore, the band gap energy (E_g_) of BiOBr is calculated to be 2.79 eV by the formula E_g_ = 1240/λ (λ is the absorption edge wavelength, 445 nm).

The separation–migration behavior of photogenerated electron–holes is tested by transient photocurrent responses and electrochemical impedance spectroscopy (EIS) of sample electrodes with illumination intervals at 20 s based on a method from a previous study [35]. As presented in Figure 5a, the photocurrent intensity of BiOBr/CNF is stronger than that of both BiOBr and CNF from vis-light irradiation, indicating that the BiOBr/CNF has a superior electron–hole separation ability. It may be attributed to the fact that BiOBr has more exposed active sites in the BiOBr/CNF hybrid and the excellent electronic conductivity of CNF. Figure 5b illustrates the EIS of samples, in which the BiOBr/CNF shows a smaller arc radius than BiOBr and CNF, implying a smaller charge transfer resistance belonging to BiOBr/CNF. Therefore, an improved photocatalytic activity of BiOBr/CNF can be obtained.

### 2.2. Adsorption and Synergistic Adsorption and Photodegradation Activities

The combined adsorption and photocatalytic performances of the samples were evaluated by monitoring their removal of TC under dark and light conditions (Figure 6a,b). As illustrated in Figure, CNF and BiOBr/CNF primarily depend on adsorption for TC removal from the solution under dark conditions. After 60 min of reaction, the concentration of TC in the solution decreased from 100 mg/L to 53.5 mg/L and 56.6 mg/L, with corresponding TC removal efficiencies of 46.5% and 43.4%. In contrast, the TC removal efficiency in the blank control group was only 5.5%. Under dark conditions, the removal efficiency of pure BiOBr on TC was only 17.2%. However, it reached 90.6% under light exposure, indicating the strong catalytic degradation ability of BiOBr on TC. Under light conditions, the removal efficiency of CNF for TC was 58.1%, showing only a 2% increase compared with dark conditions, whereas the removal efficiency of BiOBr/CNF for TC reached 94.2%, marking a 3.6% increase compared with the addition of BiOBr alone. These results indicate the effective synergistic catalytic degradation ability of the constructed BiOBr/CNF composite for TC. Moreover, the adsorption and degradation performance of tetracycline in samples under UV light exposure is shown in Figure 6c. As can be seen from Figure 6b,c, good results for the removal rate of tetracycline under visible light and UV light exposure were obtained when BiOBr/CNF was used as the catalyst. The reason for this might be that the prepared BiOBr/CNF catalyst exhibits excellent light absorption in the range of 200 nm to 445 nm (See Figure 4).

### 2.3. Analysis of Effect Factors on Tetracycline Degradation

Under light conditions, the removal efficiency of the BiOBr/CNF composite surpassed that of pure CNF and BiOBr, indicating that BiOBr, as an exceptional photocatalyst, can effectively degrade the pollutants adsorbed onto the surface of the composites. Further research was conducted to examine the factors influencing the catalytic degradation of TC (Figure 7a–c). As shown in the figure, the adsorption and photocatalytic degradation of TC using the BiOBr/CNF composite at various pH values follow the order pH 7 > pH 5 > pH 9 > pH 3. This phenomenon can be explained as follows: in an acidic environment, TC exists in a neutral or negatively charged state [36]. In addition, the surface of the BiOBr/CNF composite carries a negative charge. Thus, electrostatic repulsion occurs between the TC and the composite, reducing the possibility of contact between the two materials. At a pH of 7, the remarkable adsorption and photocatalytic degradation of TC can be attributed to favorable electrostatic interaction [37]. When the pH > 7, the TC and the surface of the composite may become positively charged, leading to electrostatic repulsion. Therefore, a pH of 7 was selected as the optimal reaction pH. As shown in Figure 7b, the initial concentration of TC significantly influences its removal. At lower concentrations of TC, the adsorption and photocatalytic degradation ability of TC are remarkable because of the sufficient number of active sites in the BiOBr/CNF composite. In addition, as the initial concentration of TC increases, the adsorption and photocatalytic degradation abilities of the composite decrease due to a reduction in the active sites. In Figure 7c, the findings indicate that the adsorption capability and photocatalytic activity are dose dependent. TC removal primarily relies on the photocatalytic behavior at low dosages and adsorption behavior at high dosages. At intermediate dosages, a substantial synergistic effect in TC degradation was observed.

### 2.4. Photocatalytic Degradation Mechanism

The free radical trapping experiment results (Figure 8) showed that, when butanol and KBrO_3_ were added, the photocatalytic removal efficiency of TC changed little, indicating that •OH and e^−^ were not the main active substances used by BiOBr to degrade TC. After adding ethylenediaminetetraacetic acid (EDTA), the degradation of TC was inhibited and the TC removal efficiency decreased by 24.4% compared to the control group, revealing that h^+^ has a certain impact on the photocatalytic degradation of TC. The photocatalytic degradation of TC by BiOBr was significantly disrupted after the addition of L-ascorbic acid (L-AC). Therefore, it can be inferred that the main active free radical for BiOBr photocatalytic degradation of TC is •O_2_^−^. Based on the characterization analysis, it can be concluded that, during the photocatalytic degradation of TC by BiOBr, hydroxyl radicals can be converted to •O_2_^−^ through the •OH → H_2_O_2_ → •O_2_^−^ pathway, playing a positive role in TC degradation [38].

Generally, the production of radicals is related to the band position of a semiconductor. Thus, the valence band (VB) and conduction band (CB) of BiOBr can be calculated using the empirical equations of E_VB_ = X − E_C_ + 0.5E_g_ and E_CB_ = E_VB_ − E_g_, where E_CB_ and E_VB_ are the CB and VB potential, X is the absolute electronegativity of the BiOBr defined as the geometric mean of the absolute electronegativity of the constituent atoms (Bi, O, and Br), and E_C_ is the energy of free electrons on the hydrogen scale (∼4.5 eV). Based on that, the E_VB_ and E_CB_ of BiOBr are estimated to be +3.34 eV and +0.55 eV. As shown in Figure 9, the generation of a large amount of •O_2_^−^ is the main reason for the improved visible light catalytic degradation performance of BiOBr composites for TC. Research has shown that Bi atoms and BiOBr semiconductors can be simultaneously excited by visible light, with semiconductor valence band electrons transitioning to the conduction band, and the electron hole pairs being excited and separated [39]. Due to the presence of O vacancies in BiOBr, it can perform electron capture well to promote the photogenerated electron–hole separation. Thus, the effective photogenerated electrons migrate to the surface of the photocatalyst. However, the potential of CB is +0.55 eV, which is not enough to reach the potential energy for •O_2_^−^ generation from O_2_ molecules. Fortunately, the CNF can not only serve as a photosensitizer to enhance visible light adsorption of BiOBr, but also can transfer electrons to the CB of BiOBr. Such transfer electrons on CNF surface could further react with the adsorbed and activated O_2_ molecules to generate •O_2_^−^ radicals. Photogenerated holes (h^+^) on the VB of BiOBr (+3.34 eV) react with activated H_2_O molecules (or OH^−^) to produce •OH radicals. These free radicals participate in the oxidation of TC, ultimately oxidizing large molecules of TC into inorganic small molecules. On the other hand, photogenerated holes (h^+^) react with activated H_2_O molecules (or OH^−^) to produce •OH radicals. These free radicals participate in the oxidation of TC, ultimately oxidizing large molecules of TC into inorganic small molecules.

### 2.5. Analysis of Intermediate Products and Photocatalytic Degradation Mechanism

To identify the intermediate products of TC during the photocatalytic reaction with the BiOBr/CNF composite, HPLC-MS was used. Distinct M/Z peaks were observed at 467, 461, 318, 202, 113, 111, 85, 61, and 57. After analysis, these peaks were attributed to the molecular ions of various intermediates, as shown in Figure 10. Furthermore, a photocatalytic degradation pathway was proposed.

## 3. Materials and Methods

### 3.1. Preparation of BiOBr

BiOBr was synthesized using the solvothermal method by dissolving 2 mmol Bi(NO_3_)_3_∙5H_2_O and 2 mmol KBr in 25 mL of ethylene glycol. After stirring for 2 h, the mixture was transferred into a 100-mL stainless steel high-pressure vessel lined with Teflon. The high-pressure vessel was heated at a self-generating pressure of 160 °C for 24 h and subsequently cooled to room temperature. The resulting precipitate was collected, washed several times with ethanol and deionized water, and dried in an oven at 80 °C. The obtained product was identified as BiOBr.

### 3.2. Preparation of BiOBr/CNF Composite Materials

Preactivation of CNF was performed by refluxing it in a dilute nitric acid solution (3 mol/L) at 100 °C for 10 h. Subsequently, the refluxed CNF was washed with pure water and dried at 60 °C for 24 h. Next, 5 g of Bi(NO_3_)_3_∙5H_2_O was dissolved in 40 mL of acetic acid, and the activated CNF was added to the solution and sonicated for 30 min. Furthermore, 0.058 g of KBr was dissolved in 20 mL of deionized water. Under vigorous stirring, the KBr aqueous solution was added dropwise to the Bi^3+^-CNF suspension, which was subsequently transferred to a stainless steel high-pressure vessel and maintained at 180 °C for 24 h. The obtained products were washed thrice with water and ethanol and then dried at 60 °C for 12 h to obtain the BiOBr/CNF catalyst. A schematic diagram of the preparation route of the BiOBr/CNF can be found in Figure 11.

### 3.3. Photocatalytic Degradation of Tetracycline

The efficiency of TC removal from the solution by CNF, BiOBr, and BiOBr/CNF under light avoidance and illumination conditions was examined using TC as the target compound. A 300-W Xenon lamp (CEL single bond S500, Beijing Zhongjiao Jinyuan Technology Co., Ltd., Beijing, China) with a light intensity of 178 mW/cm^2^ was used as the light source. Briefly, a specified amount of the sample was dispersed in a 100 mL TC solution (100 mg/L). The aforementioned solution was subjected to magnetic agitation for 100 min to achieve adsorption and desorption equilibrium. At fixed 10-min intervals, 5 mL of the reaction solution was extracted and centrifuged. The change in the TC concentration was measured using a UV–vis spectrophotometer at 350 nm. The photodegradation coefficient (η) was computed using the following Formula (1):(1)$$\eta \left(\%\right)=\left(1-\frac{C_{t}}{C_{0}}\right)\times 10\%,$$
where *C*_0_ represents the initial concentration of TC and *C_t_* represents the TC concentration at the reaction time.

### 3.4. Free Radical Trapping Experiment

BiOBr/CNF was used as the photocatalytic material. Under the condition of TC solution concentration of 100 mg/L and 100 mg/L of catalyst, L-ascorbic acid (L-AC), butanol, ethylenediaminetetraacetic acid (EDTA), and potassium bromate (KBrO_3_) with a capture agent dosage of 1 mM were selected as free radical scavengers to capture free radicals such as superoxide radicals (•O_2_^−^), hydroxyl radicals (•OH), photogenerated holes (h^+^), and photogenerated electrons (e^−^).

### 3.5. Detection Method

#### 3.5.1. Catalyst Characterization

XRD detection of the obtained material was performed using a Bruker D8 advanced diffractometer (40 kv/30 Ma, copper Kα radiation source, Berlin, Germany) supported by Serqi Technology Co., Ltd. (Beijing, China). FTIR of the sample material was conducted using an infrared spectrometer (IRA-Finity-1FTIR, 4000–400 cm^−1^) produced by Shimadzu Instrument Company (Tokyo, Japan). The morphology of the samples was examined using SEM (Fei Quanta 450, Portland, OR, USA) and TEM (Philips CM 200, Amsterdam, The Netherlands).

#### 3.5.2. Analysis of the Degradation Product

The Dinonex Ultimate 3000 UHPLC-MS (Thermo Scientific Q Exactive, Waltham, MA, USA) was employed for the analysis of intermediate products, with the following detailed test condition: Eclipse Plus C18 100 mm × 4.6 mm (3.5 μm) (Agilent, Palo Alto, CA, USA) was employed as the chromatographic column, the mobile phase comprised ultrapure water with 0.1% formic acid (phase A) and acetonitrile (phase B) at a flow rate of 0.5 mL/min with the following gradient: 0–2 min, 90% A; 10–15 min, 10% A; 16–20 min, 90% A. The column temperature was maintained at 30 °C, and the injection volume was 20 μL. In addition, mass spectra were recorded in negative ionization mode over the full scan range of 30–500 *m*/*z*.

## 4. Conclusions

In this research, a novel photocatalyst made of BiOBr combined with cellulose nanofibers (BiOBr/CNF) was successfully synthesized using a solvothermal approach. The BiOBr/CNF exhibited high catalytic degradation activity for TC under visible light across a wide pH range. Free radical trapping experiments demonstrated that •O_2_^−^ and •OH free radicals were the main contributors to the degradation process of TC using BiOBr/CNF under visible light. The TC degradation pathways were proposed based on the emerging intermediates detected during the degradation process by LC-MS. Our study clarifies the correlation between the microstructure of BiOBr/CNF composite catalysts and their improved photocatalytic performance, providing an innovative approach for efficiently eliminating water pollutants.

## Figures and Tables

**Figure 1 molecules-29-03253-f001:**
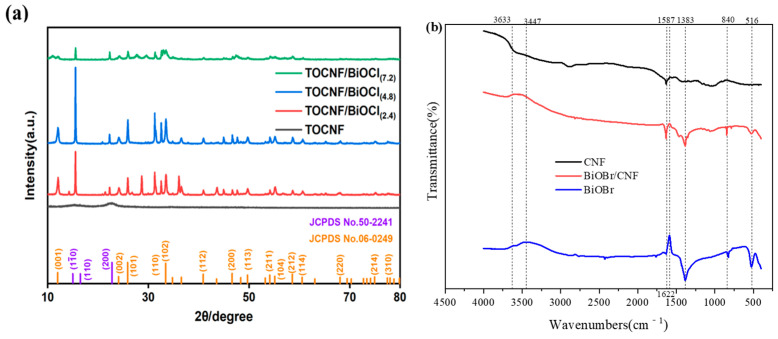
(**a**) X-ray diffraction patterns and (**b**) Fourier transform infrared spectra of the prepared samples.

**Figure 2 molecules-29-03253-f002:**
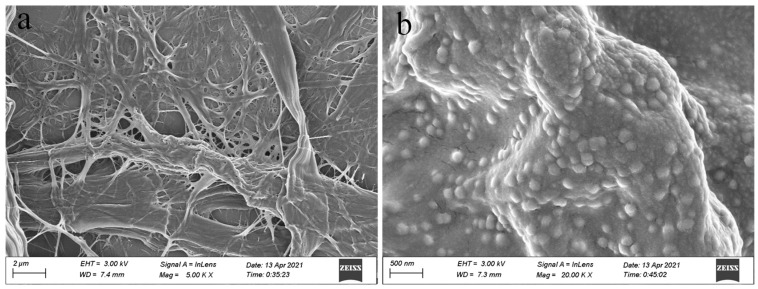
Scanning electron microscopy images of (**a**) cellulose nanofibers (CNF), (**b**) BiOBr/CNF.

**Figure 3 molecules-29-03253-f003:**
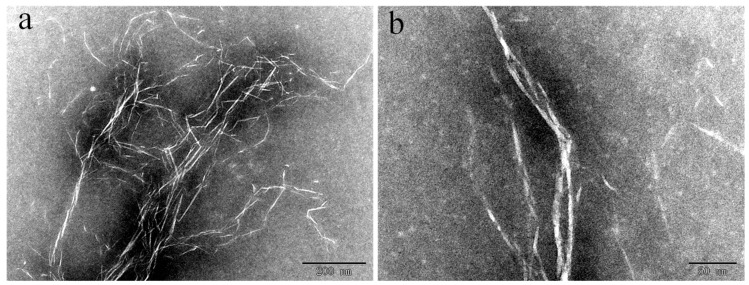
Transmission electron microscopy images of (**a**) CNF and (**b**) BiOBr/CNF.

**Figure 4 molecules-29-03253-f004:**
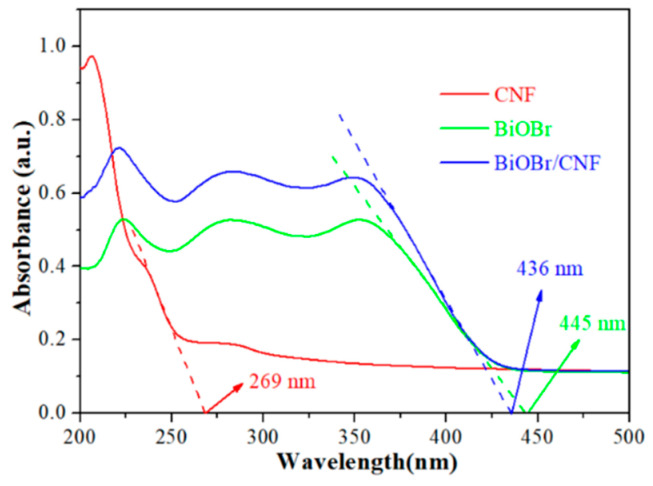
UV–vis diffused reflectance spectra of samples.

**Figure 5 molecules-29-03253-f005:**
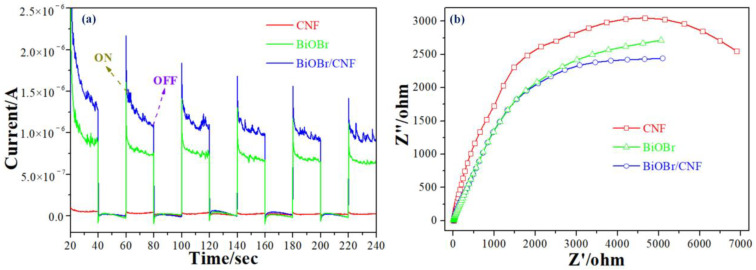
The I-T curves (**a**) and the EIS response (**b**) of sample.

**Figure 6 molecules-29-03253-f006:**
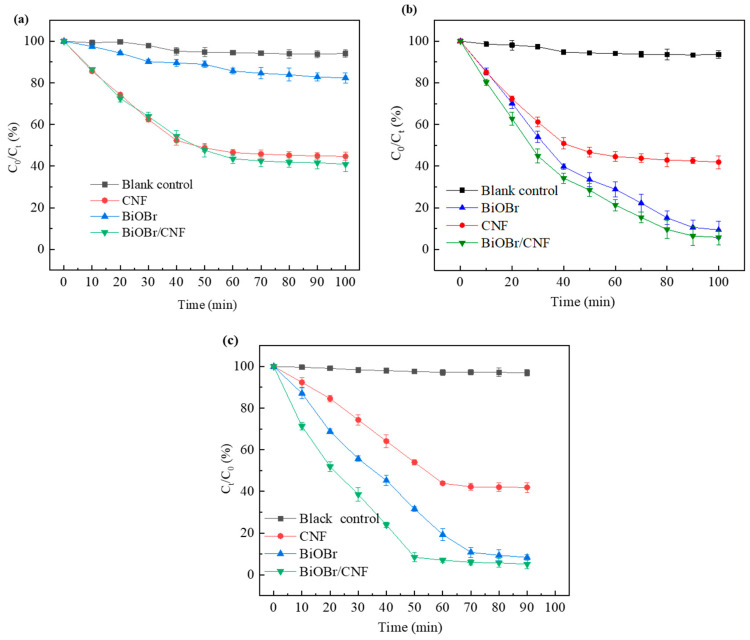
Adsorption and degradation performance of tetracycline in various samples under (**a**) light avoidance, (**b**) visible light exposure, and (**c**) UV light exposure.

**Figure 7 molecules-29-03253-f007:**
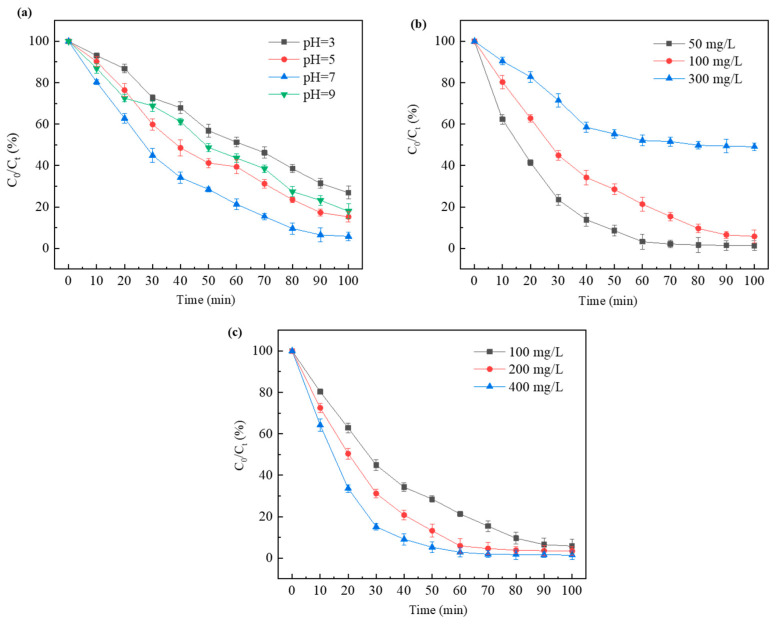
Effects of (**a**) pH, (**b**) initial TC concentration on removal efficiency, and (**c**) catalyst dosage.

**Figure 8 molecules-29-03253-f008:**
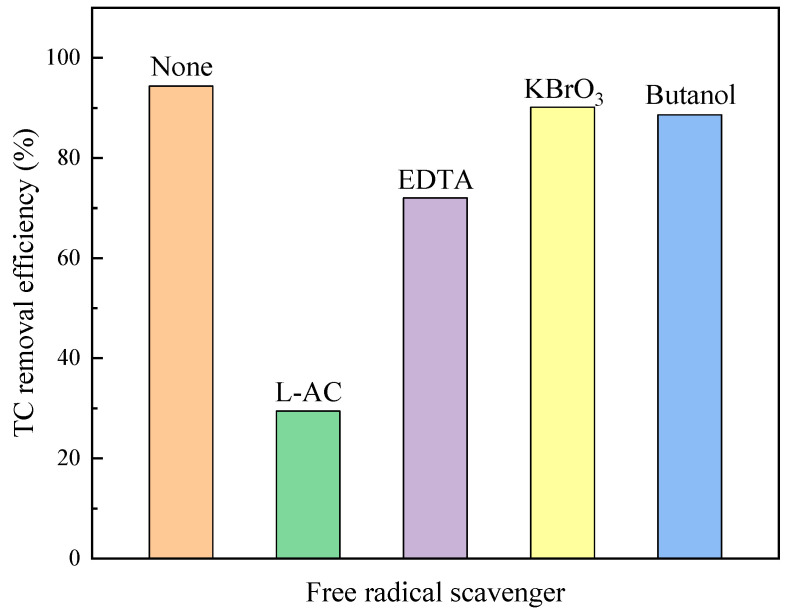
Free radical trapping with different free radical scavenger.

**Figure 9 molecules-29-03253-f009:**
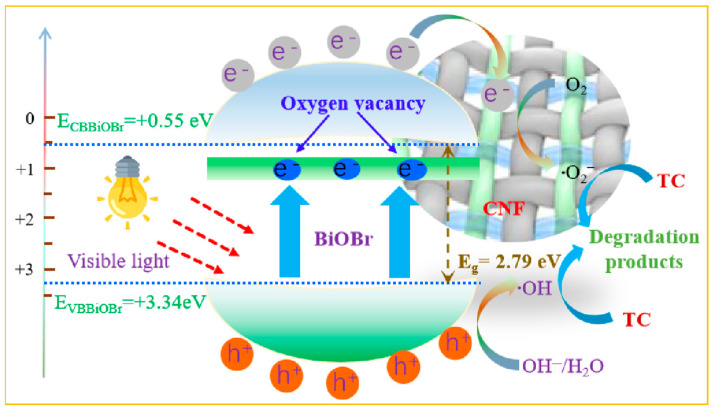
Photocatalytic degradation mechanism of tetracycline by CNF/BiOBr.

**Figure 10 molecules-29-03253-f010:**
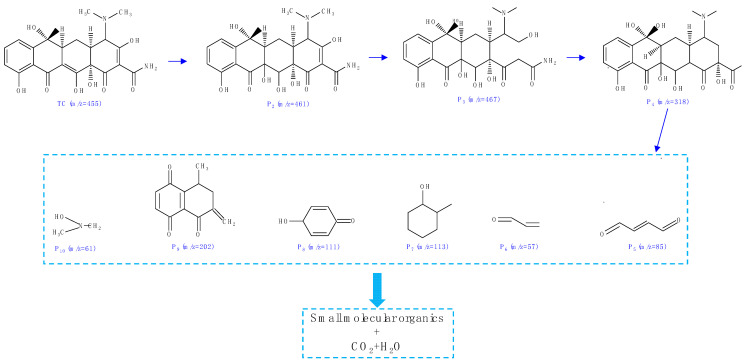
Possible degradation pathway of the CNF/BiOBr composite.

**Figure 11 molecules-29-03253-f011:**
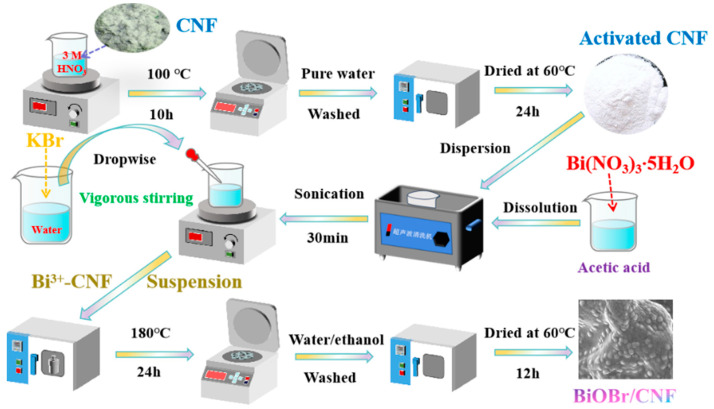
Schematic diagram of the preparation route of the BiOBr/CNF.

## Data Availability

Data will be made available on request.

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
