# Peer review of "Construction of a Wood Nanofiber–Bismuth Halide Photocatalyst and Catalytic Degradation Performance of Tetracycline from Aqueous Solutions"

_molecules, 2024, doi:10.3390/molecules29143253_

Round 1
Reviewer 1 Report
Comments and Suggestions for Authors
The manuscript, which details the construction of a Wood Nanofiber–Bismuth Halide Photocatalyst and its catalytic degradation performance of Tetracycline from aqueous solutions, can be approved for publication following significant revisions.
1) The authors should clearly highlight the novelty of their work in the introduction section. This will help readers understand the unique contributions of the study right from the beginning. Including this information is essential for setting the context and framing the research's importance.
2) The XRD file reference from the JCPDS database should be included in Figure 1a.
3) In the context of photocatalytic degradation, why have you chosen to work with high concentrations (50 to 300 mg/L)? Could you please elaborate on the rationale behind selecting these specific concentration ranges for your experiments?
4) The authors should determine the band gap energy of the prepared material, referencing the methods described in https://doi.org/10.1016/j.seppur.2021.119399 and https://doi.org/10.1016/j.colsurfa.2023.131509 . This calculation is crucial for understanding the electronic properties of the material in question.
5) Could you please provide a detailed explanation of the photocatalytic degradation mechanism, focusing on the calculated conduction and valence bands?
6) can you explain the role of the internal electric field (IEF) in facilitating the separation and migration of photoexcited electron-hole pairs within the BiOX structures mentioned in the study?*
7) How does the photocatalytic degradation performance of the BiOBr/CNF composites under visible light compare to their performance under ultraviolet light?
Author Response
Please see the attachment of Responses to Reviewers Comments-1

Reviewer 2 Report
Comments and Suggestions for Authors
The paper presents interesting and original results on the use of a new photocatalyst for water purification. There are some remarks:
1. Authors should indicate the purpose and novelty of the work in the introduction
2. The main conclusions of the work should be summarized in the conclusion section (now there is no conclusion)
3. Experimental errors should be indicted in the figures with kinetic experiments
4. What is average particle size for the prepared materials?
Author Response
Please see the attachment of Responses to Reviewers Comments-2

Reviewer 3 Report
Comments and Suggestions for Authors
The combination of BiOBr and cellulose into composites can promote the synergistic adsorption and photodegradation of TC, much work has been done, this work will be more better after great improvement.
1. It is better to provied Standard card of the samples in the Figure 1. (a) X-ray diffraction patterns and, and mark the peaks in (b) Fourier transform infrared spectra of the prepared samples.
2. It is better to provide the CB, VB and Eg of the samples, besides, more investigation about the charge transfer between BiOBr and CNF should be conducted to help the reader to understand the advantage of this composite.
3. It is better to provide a schematic diagram of the preparation route of the samples to understand this work.
4. It is better to investigate the electrochemically active surface area of the sample, that can employ to show the active site of this catalyst, ref. https://www.sciencedirect.com/science/article/abs/pii/S1005030223005807
5. In Figure 7, what is the role of O vacancy and CNF?
Author Response
Please see the attachment of Responses to Reviewers Comments-3

Round 2
Reviewer 1 Report
Comments and Suggestions for Authors
No comments
Reviewer 2 Report
Comments and Suggestions for Authors
Thanks for corrections